



# Novel estimation of aerosol processes with particle size distribution measurements: a case study with TOMAS algorithm

Dana L. McGuffin[1], Yuanlong Huang[2], Richard C. Flagan[2], Tuukka Petäjä[3], B. Erik Ydstie[4], Peter J. Adams[5]

[1]Lawrence Livermore National Laboratory, Livermore, CA, 94550, United States
[2]Division of Chemistry and Chemical Engineering, California Institute of Technology, Pasadena, 91125, United States
[3]Institute for Atmospheric and Earth System Research (INAR) / Physics, Faculty of Science, University of Helsinki, Finland
[4]Department of Chemical Engineering, Carnegie Mellon University, Pittsburgh, 15213, United States
[5]Center for Atmospheric Particle Studies, Carnegie Mellon University, Pittsburgh, 15213, United States

*Correspondence to*: Dana L. McGuffin (dana.lynn.mcguffin@gmail.com)

**Abstract.** Atmospheric aerosol microphysical processes are a significant source of uncertainty in predicting climate change. Specifically, aerosol nucleation, emissions, and growth rates, which are simulated in chemical transport models to predict the particle size distribution, are not understood well. However, long-term size distribution measurements made at several ground-based sites across Europe implicitly contain information about the processes that created those size distributions. This work aims to extract that information by developing and applying an inverse technique to constrain aerosol emissions as well as nucleation and growth rates based on hourly size distribution measurements. We developed an inverse method based upon process control theory into an online estimation technique to scale aerosol emissions, growth, and nucleation so that the model-measurement bias in three measured aerosol properties exponentially decays. The properties, which are calculated from the measured and predicted size distributions, used to constrain aerosol nucleation, emission, and growth rates are the number of particles with diameter between 3 nm and 6 nm, the number with diameter greater than 10 nm, and the total dry volume of aerosol ($N_{3-6}$, $N_{10}$, $V_{dry}$), respectively. In this paper, we focus on developing and applying the estimation methodology in a zero-dimensional "box" model as a proof-of-concept before applying it to a three-dimensional simulation in subsequent work. The methodology is first tested on a dataset of synthetic and perfect measurements that span diverse environments in which the true particle emissions, growth, and nucleation rates are known. The inverse technique accurately estimates the aerosol microphysical process rates with an average and maximum error of 2% and 13%, respectively. Next, we investigate the effect that measurement noise has on the estimated rates. The method is robust to typical instrument noise in the aerosol properties as there is a negligible increase in bias of the estimated process rates. Finally, the methodology is applied to long-term datasets of in-situ size distribution measurements in Western Europe from May 2006 through June 2007. At Melpitz, Germany and Hyytiälä, Finland, the average diurnal profiles of estimated 3 nm particle formation rates are reasonable, having peaks near noon local time with average peak values of 1 and $0.15~\text{cm}^{-3}\text{s}^{-1}$, respectively. The normalized absolute error in estimated $N_{3-6}$, $N_{10}$, and $V_{dry}$ at three European measurement sites is less than 15%, showing that the estimation framework developed here has potential to decrease model-measurement bias while constraining uncertain aerosol microphysical processes.

## 1. Introduction

Atmospheric aerosols scatter and absorb incoming solar radiation (Cubasch et al., 2013). They also provide sites for cloud droplet formation, known as cloud condensation nuclei (CCN) (Twomey, 1974). Through the latter process, they contribute to the aerosol indirect effect in which aerosols perturb the CCN concentrations, affecting the Earth's energy balance by altering





cloud properties such as cloud coverage, reflectivity, and lifetime (Lohmann and Feichter, 2005). The indirect effect is the most uncertain mechanism of global radiative forcing (Stocker et al., 2013). Given the importance of aerosol indirect radiative forcing, considerable effort has been spent developing chemical transport models (CTMs) that predict CCN concentration fields (Adams and Seinfeld, 2002; Easter, 2004; Kajino and Kondo, 2011; von Salzen et al., 2000; Spracklen et al., 2005, 2010;

Wilson et al., 2001; Yu and Luo, 2009; Zhang et al., 2004). CTMs simulate atmospheric processes and aerosol microphysics driven by meteorological fields.

The major sources of uncertainty in using a CTM to predict CCN include uncertainties in the rates of the following processes: particle formation, emissions, and growth (Carslaw et al., 2013; Pierce and Adams, 2009b; Spracklen et al., 2008, 2011; Westervelt et al., 2014). New particle formation, or nucleation, is the formation of thermodynamically stable clusters

near 1 nm in diameter from condensable vapours. The general understanding of nucleation is still an open challenge (Adams et al., 2013). Particle emissions can be estimated from data such as emission factors and activity levels of each emission source for different classes of sources. However, this data is less commonly tabulated for number concentration emitted as opposed to mass emitted, and measurements of the emitted particle size distributions are sparse. Particles grow in size due to the condensation of sulphuric acid and oxidized volatile organic compounds (VOCs). An uncertain amount of VOCs are emitted

from biomass burning, anthropogenic sources, and the biosphere (Folberth et al., 2006). Then, sulphuric acid and VOCs form secondary organic aerosol (SOA) (e.g. Kerminen et al., 2018; Kulmala et al., 2014; Shrivastava et al., 2017), but the SOA yield from VOCs is also uncertain.

Predicted CCN concentrations can be improved either by nudging the concentration fields themselves, or by estimating the particle formation, emissions, and growth rates that largely control CCN concentrations. Estimating the key uncertain aerosol

processes is preferred to directly estimating CCN concentrations because it provides insight to the underlying model biases (Benedetti et al., 2018). A systematic way to estimate uncertain atmospheric processes is to use data assimilation or inverse modelling techniques that employ a combination of CTMs and field measurements.

Long-term observations of particle size distributions are available from large measurement networks such as the European Supersites for Atmospheric Aerosol Research (EUSAAR) (Asmi et al., 2011), German Aerosol Ultrafine Network (GUAN)

(Birmili et al., 2015), and Global Atmosphere Watch World Data Centre for Aerosols (GAW-WDCA) (www.gaw-wdca.org). Observed size distributions contain intrinsic information about the aerosol processes that created those size distributions. Previous work focuses on extracting particle formation and growth rates from size distributions observed at a measurement site (Kerminen et al., 2018). These studies utilize methods developed by Kulmala et al. (2012) and references therein to determine aerosol process rates during new particle formation (NPF) events. However, there are several limiting assumptions

in the proposed methodology, including: the coagulation sink and growth rate are assumed to be constant across particle sizes as is the nucleation rate during the NPF event; moreover, no sources of particles between 3 nm and 25 nm are considered besides NPF. A method that uses an aerosol microphysics model or a CTM with aerosol microphysics could account for the limiting assumptions to enable the estimation of aerosol processes outside of as well as during NPF events. The goal of this





paper is to propose a computationally efficient method that integrates size distribution measurements with an atmospheric aerosol model that improves model accuracy by inferring aerosol process rates.

Previous studies used size distribution measurements from smog chamber experiments with 0D ("box") models simulating the aerosol general dynamic equation (GDE) to estimate uncertain terms in the GDE (Pierce et al., 2008; Verheggen and

Mozurkewich, 2006). These inverse models estimate processes such as nucleation, growth, and chamber wall loss by minimizing the model-measurement bias. The minimization procedure involves iteratively fitting the model to the measurements, and requires knowledge of the sensitivity of the size distribution to the uncertain processes for each iteration. However, this method does not guarantee that the optimal process rates will be discovered within a low number of iterations, making the inverse model potentially computationally expensive when applied in the context of a three-dimensional model.

Furthermore, previous inverse modelling studies with 3D CTMs have focused on estimating emission rates of gases, such as methane, ammonia, sulfur oxides, and nitrogen oxides (Bergamaschi et al., 2010; Gilliland et al., 2003; Hein et al., 1997; Henze et al., 2008; Houweling et al., 1999) from in-situ measurements.  Other investigations have estimated aerosol emissions from remote aerosol optical depth measurements (Chen et al., 2019; Dubovik et al., 2008; Escribano et al., 2016, 2017; Huneeus et al., 2012, 2013; Wang et al., 2012; Xu et al., 2013; Zhang et al., 2015, 2005) or from in-situ observations (Viskari

et al., 2012b, 2012a). These previous researchers used various inverse modelling techniques to solve their proposed problem, including a Kalman filter and a four-dimensional variational (4D-var) method with an adjoint model. These techniques are not ideal to estimate aerosol microphysical processes for three reasons: 1) complexity in the relationship between ambient size distribution and the emissions and aerosol formation and growth, 2) significant computational cost in optimizing multiple (potentially spatially distributed) parameters, and 3) the adjoint model used with 4D-var becomes obsolete when the 3D CTM

is updated.

In this study, we utilize estimation techniques from the field of nonlinear process control to address disadvantages in the current inverse modelling techniques. Understanding a complex process is vital when controlling it to a set point or goal, so adaptive controllers utilize online estimation algorithms to improve the controller's internal model with data. We apply ideas from inventory control and passive systems theory (Farschman et al., 1998; Ydstie, 2002) to formulate an estimation algorithm

for aerosol microphysics. Inventory control uses a set of variables, called inventories, to define the overall performance of the inverse model. These ideas have been used to control the float-glass process (Ydstie and Jiao, 2006), a pressure tank (Li et al., 2010), and production of solar-grade silicon (Balaji et al., 2010). The same theories have been used to estimate chemical kinetics and heat of reaction (Zhao and Ydstie, 2018). An estimation technique based on inventory control is attractive because it is developed for complex and nonlinear systems, does not require significant computational cost, and is flexible to model

updates because of the algorithm's high-level perspective. By this, we mean that the algorithm needs to know the net rates of certain processes but is insensitive to the details of how those rates are calculated. In coding terms, the process rates can be estimated by looking at changes before and after the corresponding subroutine is called and is robust to changes in the subroutine itself, setting it apart from adjoint methods.



In this work, we aim to design an inverse modelling technique from nonlinear process control theory that can incorporate size distribution measurements with a 3D CTM; however, as a first development step, we limit the estimation algorithm to a box model. Our objective is to input size distribution measurements to the inverse model in order to estimate uncertain aerosol processes simulated in a box model: particle formation, emissions, and SOA production rates. We will first describe the inverse

5 model and how it is designed to estimate particle formation, emissions, and growth. Then, we will validate the method on sets of synthetic measurements relevant to conditions in a 3D CTM. Next, we will assess the effect of instrument noise on the estimates by corrupting the synthetic measurements with a realistic noise signal. Finally, we will test the inverse model on realistic field data by estimating time-varying particle formation, emissions, and growth at three measurement sites: San Pietro Capofiume, Italy; Melpitz; Germany, and Hyytiälä, Finland. While the ultimate goal of this work is to deploy the inverse

10 method in a 3D CTM, all of the steps presented here are proof-of-concept work in a zero-dimensional atmospheric "box" model.

## 2. Inverse Modelling Methods

Inverse models use the observed model output to estimate a set of control variables such that the predicted model output matches the observations as nearly as possible. Common control variables estimated in the atmospheric inverse modelling/data

15 assimilation fields include emission fluxes and mixing ratios. A disadvantage of using mixing ratios as control variables is that it does not address underlying errors in processes that are responsible for the mismatch between the model and observations. While model-predicted mixing ratios are improved, little insight is gathered into causes of the errors. In this work, we consider control variables that are scaling factors applied to three highly uncertain but important aerosol processes: particle nucleation, emissions, and growth. Since these processes control the evolution of the measured properties, we anticipate improved

20 performance over mixing-ratio control variables.

### 2.1. TOMAS Model

The TwO-Moment Aerosol Sectional (TOMAS) algorithm simulates both the discretized mass and number size distributions. In this work, we utilize a zero-dimensional version of TOMAS as our "box model". The algorithm was originally described by Adams and Seinfeld (2002) based on numerical methods for simulating cloud droplet microphysics originally proposed by

25 Tzivion et al. (1987, 1989). The code has been updated several times since the original release (Pierce and Adams, 2009a; Trivitayanurak et al., 2008; Westervelt et al., 2013). The TOMAS box model used here simulates sulphuric acid, ammonia, and sulphur dioxide vapors as well as five particle species: sulphate, ammonium, sea-salt, organic carbon, water. The discretized size distribution includes 43 size sections, defined by particle mass, that are logarithmically-spaced by a factor of two. The smallest particle is $1.22 \cdot 10^{-25}$ kg dry aerosol mass per particle, resulting in a size distribution that spans particle dry





diameters from roughly 0.5 nm to 10 $\mu$m, based on a typical particle density of 1.8 g cm$^{-3}$. TOMAS calculates the density online based on the current composition of a given size bin.

In this work we utilize a simplified version of TOMAS that was described by Westervelt et al. (2013). The simulated microphysics include nucleation, coagulation, condensation, wet deposition, size-resolved dry deposition, and emissions. The
model incorporates a combination of binary and ternary nucleation mechanisms (Napari et al., 2002; Vehkamäki et al., 2002) in which the ternary parameterization allows calculation of the rate of formation of new particles 3 nm in diameter if the concentration of ammonia gas exceeds 0.1 ppt. Condensation of sulphuric acid vapour and VOCs follow a kinetic-scheme in which the vapour condenses to Fuchs-corrected surface area (Riipinen et al., 2011). The rate of aerosol mass accumulation due to condensation of VOCs is defined here as the production rate of secondary organic aerosol (SOA), in which the combined
processes of VOC emission, chemistry, and condensation result in a total SOA production rate. Sulphuric acid vapor is assumed to be in pseudo-steady-state between aerosol nucleation, growth, and its photochemical production from sulphur dioxide (Pierce and Adams, 2009a). Organic carbon aerosol, sulphur dioxide, and ammonia are emitted at a constant flux in which the primary organic aerosol (POA) emissions are based on measured size distributions of particles emitted from heavy- and light-duty vehicles (Ban-Weiss et al., 2010). Sea spray emissions are not considered here since we are simulating a continental
measurement station. Therefore, sea-salt does not contribute to the simulated aerosol composition. POA emission, SOA production, and particle nucleation rates will be adjusted based on the measurements, so the *a priori* rates are not significant here.

Particle sinks, such as wet deposition and transport, are simplified as a single first-order loss of particles, with a time constant of 12 hours, which is much faster than a time constant for depositional losses only. This high loss rate is important
for our application of the box model to simulate field measurements. A weakness of any application of a box model to ambient data is that advection, convection, and dilution are not simulated explicitly. The microphysical processes constrained here are all aerosol sources, so a large sink will allow the box model to match measurements that are rapidly decreasing, e.g., due to an inflow of cleaner air. Dry deposition is calculated with a resistance-in-series approach (Zhang et al., 2001).

## 2.2. Parameter Estimation Technique

To perform the inverse modelling technique, we adjust the TOMAS box model by introducing three time-varying scaling factors as the control variables that we want to estimate. Then, we will estimate the control variables from moments of the size distribution that are sensitive to the uncertain processes. The estimation method used here requires us to define "inventory variables", which are measureable quantities that are additive, positive, and continuously differentiable (McGuffin et al., 2019b). The observed and predicted size distributions are projected to the inventory variables ($y_k$) at time $t_k$, which are used
to estimate the set of scaling factors input to TOMAS, as shown in Figure 1. Here we choose to employ as inventory variables: 1) the particle concentration between 3 nm and 6 nm ($N_{3-6}$), 2) the number concentration of particles greater than 10 nm ($N_{10}$), 3) and the dry aerosol volume concentration ($V_{dry}$) ($y_k = [N_{3-6} \quad N_{10} \quad V_{dry}]^T$). These three variables are strongly sensitive





to the uncertain processes: nucleation, emissions, and growth, respectively. The rates of change of the inventory variables depend on the scaling factors at time $t_k$ ($\mu_k$) such that

$$\frac{dy_k}{dt} = f_k + G_k \mu_k \tag{1}$$

where $G_k$ is an array of the sensitivity of each inventory variable with respect to each uncertain process and $f_k$ is the vector of

remaining terms in the inventory dynamics not directly dependent on the uncertain processes. Conceptually, Equation (1) can be understood as a balance equation for the inventory variable, which is derived from the general dynamic equation for aerosol microphysics. For example, when $y_k$ corresponds to $N_{10}$, the terms on the right consist of the rates of all the processes that add or remove particles from this size range. For $N_{10}$, this would include the emissions rates of any particles larger than 10 nm and growth of smaller particles to larger than 10 nm sizes in $G_k$, as well as formation or loss of particles in this size range by

coagulation (in $f_k$). Theoretically, they could be derived by a suitable integration of the aerosol general dynamic equation. In practice, in the code, $G_k$ and $\frac{dy_k}{dt}$ are easily calculated online based on a backward finite difference scheme since the model equations are solved using a forward explicit Euler technique. Then, $f_k$ is determined from Equation (1) based on the nominal scaling factor $\mu_k$ used to determine $G_k$. Because these rates are determined by finite difference, i.e., by saving model parameters before and after relevant subroutines are called, this approach is fairly modular and robust with respect to changes

in the aerosol microphysics. Internal details of subroutines can change so long as the estimator is able to compare the model state before and after the subroutine call.

The parameter estimation technique was described in detail by McGuffin et al. (2019b); it was previously used to estimate sea spray emissions in a 3D CTM (McGuffin et al., 2019a). We will give a brief summary of the parameter estimation technique here. Instead of using a least-square regression or the analytical maximum a posteriori solution, as other parameter estimates

are generated (i.e., variational data assimilation or Bayesian Inference techniques), we update the parameter estimate so that the model-measurement error exponentially decays, as shown in Equation (2). In this case, the error is defined based on the chosen inventory variables ($y_k$). The parameter estimate is calculated by solving the system of linear equations

$$f_k + G_k \hat{\mu}_k - \frac{dy_k^{obs}}{dt} = -K_c \left( \hat{y}_k - y_k^{obs} \right) \tag{2}$$

where $\hat{y}_k$ is the model-predicted vector of inventory variables at time $t_k$, $y_k^{obs}$ is the observed vector of inventory variables,

$\frac{dy_k^{obs}}{dt}$ is the observed derivative of the vector of inventory variables, $\hat{\mu}_k$ is the vector of estimated scaling factors, and $K_c$ is a positive-definite diagonal array that determines the error exponential decay rate. $K_c$ is analogous to a Kalman gain that weights the model-measurement bias. The gain has dimensions of frequency, i.e., inverse time. The gain determines how quickly the estimator forces the model to match the observations. High values of the gain accelerate convergence to observations. In contrast, decreasing the gain tends to weight the model more heavily than observations, increasing the time

required for the model predictions to converge to observations, so that the model does not follow the observations with high time resolution; the estimator will, however, reduce systematic biases if given sufficient time. This property of $K_c$ allows the user to avoid noisy measurements corrupting the parameter estimates. The left-hand side of Equation (2) represents the rate



of change of the model-measurement error, which is invertible for the control variables $\hat{\mu}_k$. The terms $f_k$ and $G_k$ can be easily calculated online as described above.

Because we require instantaneous sensitivity and model dynamics ($G_k$ and $f_k$), we must run the TOMAS algorithm twice for each time step: 1) to determine $f_k$ and $G_k$ from the nominal $\mu_k$, and 2) to move the simulation forward in time based on the
estimated $\hat{\mu}_k$. The solution to the parameter estimation $\hat{\mu}_k$ is implemented in the model in the second step to affect the complete number and mass size distributions.

There are two main drawbacks to the parameter estimation technique utilized here. First, we require knowledge of the derivative of the observations, which may include noise from differentiation. Unlike noise in the observations, noise in the derivatives cannot be dampened by adjusting the tuning parameter $K_c$. However, we can smooth $\frac{dy^{obs}}{dt}$ to remove any
differentiation noise with filtering techniques, such as the Savitzky-Golay filter. Second, $G_k$ must have full rank and it should be well-conditioned to solve Equation (2) for the scaling factors. If the sensitivity array is ill-conditioned, we cannot accurately solve the system of equations for the scaling factors to estimate the three process rates. Physically, this corresponds to the situation in which the measured inventory variables do not unambiguously constrain the process rates, i.e. several sets of process rates adequately satisfy the measured constraints. We determine if the system is ill-conditioned based on the condition
number ($\kappa$) of the relative sensitivity array (RSA), which is the element-wise product of the sensitivity array and the transpose of its inverse

$$RSA(t_k) = G_k \circledast (G_k^{-1})^T \qquad (3)$$
$$\kappa(t_k) = cond\big( RSA(t_k) \big).$$

Here, the condition number is calculated as the ratio between the maximum and minimum eigenvalues of the square matrix,
$RSA$, at time $t_k$.

Since the scaling factors are allowed to vary temporally, the estimated scaling factors are specific to the model and its *a priori* particle formation, emission, and growth rates. The scaling factors do not have any inherent physical meaning. Additionally, the estimated process rates cannot simply be reconstructed from the *a priori* rate and the estimated scaling factor since aerosol processes can be dependent on the state of the atmosphere, i.e., particle formation and growth depend on aerosol
surface area. For all these reasons, instead of analysing the estimated scaling factors, we will look at the estimated aerosol process rates.

### 2.3. Particle Size Distribution Observations

The particle size distribution was observed from May 2006 through June 2007 at three rural locations: San Pietro Capofiume, Italy (SPC); Melpitz, Germany (MPZ); and Station for Measuring Ecosystem – Atmosphere Relations II (Hari and Kulmala,
2005) site in Hyytiälä, Finland (HYY). All three measurement sites use twin-Differential Mobility Particle Sizer (DMPS) instruments to observe the ambient size distribution with particle diameters ranging from 3 nm to various upper size limits.





The largest particle diameter measured is 0.6 µm, 0.8 µm, and 1 µm at SPC, MPZ, and HYY, respectively. The experimental setup at SPC and HYY was described by Aalto et al. (2001), and Birmili et al. (1999) described the setup at MPZ.

Random noise in the measured inventory variables could corrupt the estimated scaling factors. Instead of directly inputting the observed inventory variables, we smooth the observations and calculate their derivatives with a Savitzky-Golay filter

(Savitzky and Golay, 1964). The filter fits a polynomial of a predetermined degree to the dataset over a time horizon that is also predetermined. The filtered value then is taken as the value of the polynomial at the midpoint of the time horizon. Additionally, the rate of change of a dataset is determined by differentiating the fitted polynomial at the midpoint. The method is computationally efficient since the there is an analytical solution to the best-fit polynomial coefficients (Savitzky and Golay, 1964).

In this work, we use the Savitzky-Golay filter of degree one so that we perform a moving horizon average on the raw measured inventory variables. Then, we use finite differences on the filtered data to calculate the derivative of the measurements. The measurement derivatives and hourly filtered measurements are used to linearly interpolate the measurements to a frequency of 5 minutes, which is the model time step.

## 3.   Validation of Inverse Modelling Technique

To evaluate the inverse modelling technique, we estimate particle formation, emissions, and growth rates based on simulated inventory variables, or "synthetic measurements". The synthetic measurements are from the TOMAS box model itself with scaling factor inputs ($\mu^*$); the aerosol formation, emissions, and growth rates simulated are the so-called "true" rates. Then, TOMAS is run with the initial scaling factors at their nominal values ($\mu \neq \mu^*$), but the inverse modelling technique adapts the scaling factors so the online model prediction matches the synthetic measurements. This configuration tests whether the

estimation technique can recover the "true" process rates starting from a biased *a priori* model. This represents a best-case, proof-of-concept because the synthetic measurements have no measurement noise and because the only errors in the *a priori* model are in the processes to be tuned. Nevertheless, it demonstrates the viability and potential performance of the theoretical approach.

Figure 2 shows how the inverse modelling approach performs for a week-long simulation in which the original model

under-predicted aerosol mass (dry aerosol volume) as well as $N_{10}$, and over-predicted the nucleation mode number concentration ($N_{3-6}$). Figures 2 (a), (c), and (e) each show the *a priori* prediction if the scaling factors are not adjusted, the synthetic measurements, and the estimated inventory variables for $N_{3-6}$, $N_{10}$, and $V_{dry}$, respectively. Figure 2 (b), (d), and (f) show the process rates for the *a priori*, true values that determine the synthetic measurements, and the estimated rates of aerosol nucleation, emission, and SOA production, respectively. The first 12 hours of the inverse modelling simulation is considered

as model spin-up, so the parameter estimation starts adjusting the scaling factors at $t = 12\ hours$. We then evaluate the performance of the estimator starting at 24 hours until the end of the simulation. The percent error between the average estimated and true value normalized by the average true value between day 1 and 7 for the aerosol measurements and rates is





shown in each plot of Figure 2. The method works very well in this case since the average bias for each measurement and aerosol process rate is less than 1% and 2%, respectively.

Since the objective is to design an inverse technique that is robust enough to apply in a global 3D CTM, we will repeat the above method for a set of 27 scenarios that span the range of uncertain process rates. We explore different particle formation,

emissions, and SOA production rates that span approximately $0.001 - 300\ cm^{-3}\ s^{-1}$, $580 - 2200\ cm^{-3}\ hr^{-1}$, and $1 - 38\ \mu g\ m^{-3}\ day^{-1}$, respectively. A weakness of this estimation method occurs when the sensitivity matrix ($G_k$) is ill conditioned. Out of the 27 scenarios investigated here, four sets of synthetic measurements are deemed ill-conditioned as they were generated from a system in which the condition number, calculated with Equation (3), is greater than 1.2 on average. In these cases, the condition number is near or equal to one for the majority of the simulation except for a small number of time

steps in which the condition number spikes much higher, leading to an average condition number between 1.2 and 2.3. Common features among these ill-conditioned scenarios are very high nucleation rates coupled with low SOA production and low emission rates. Out of the four "ill-conditioned" scenarios, three scenarios were physically unrealistic in that they paired extremely high formation and extremely low growth rates – unlikely because the photochemistry that leads to nucleation events also tends to lead to high rates of condensational growth. Figure 3 shows the average estimate against the average measured

inventory variables for all 27 scenarios. The only synthetic measurements that were not estimated accurately are ill-conditioned and unrealistic scenarios, which are denoted by the open triangle markers. The three inventory variables are estimated with an average error less than 11% for the 23 well-conditioned scenarios and the one physically realistic ill-conditioned scenario.

Similarly, Figure 4 shows that the estimated nucleation, emissions, and growth rates are accurately estimated if the synthetic measurements were well-conditioned. Particle formation only directly affects $N_{3-6}$, which is significantly more sensitive to

nucleation than emissions or growth, so we see that the nucleation rate is estimated accurately independent of the condition number. On the other hand, accuracy of the estimated emissions and SOA production rates are strongly dependent on whether the true system is well-conditioned. Particle formation, emissions, and SOA production are each estimated with an average error less than 13% in the 23 well-conditioned scenarios. In the worst-case scenario (the ill-conditioned but realistic scenario), aerosol nucleation, emissions, and SOA production rates are estimated with errors of 4%, 38%, and 9% on average,

respectively.

Although the inverse modelling technique in general estimates the correct inventory variables and aerosol process rates, we also wish to investigate whether the estimated size distribution will match the true size distribution. Accurately simulating the size distribution is very important to correctly predict the effect that aerosols have on climate. Figure 5 shows that the average estimated size distribution based on the inventory variables matches the average size distribution of the synthetic

measurements generated from an intermediate set of particle formation, emissions, and SOA production rates. For the 23 well-conditioned scenarios with low bias in the estimated aerosol process rates, the estimated size distribution similarly closely matches the true size distribution.



### 3.1. Effect of Measurement Error on Estimates

The estimation technique performs very well when utilizing "perfect measurements" ($y^*$) that are not corrupted by any measurement noise or errors, but this is not a realistic scenario. We add noise to the 23 well-conditioned synthetic measurements to understand how measurement uncertainty will affect the estimated process rates. The noise added to each
inventory variable is sampled, as random numbers, from a Gaussian probability distribution function with zero-mean and a standard deviation of the approximated uncertainty in the respective inventory variable. The same set of random numbers are sampled for each scenario so that the noise signals in each of the 23 scenarios only vary in magnitude but their temporal profiles are the same.

We calculated the uncertainty in the size distribution from an instrument model described in Appendix A using the
operating parameters of the DMPS operated at San Pietro Capofiume. The true size distribution is input to the instrument model to determine the size distribution uncertainty, which assumes Poisson counting statistics for each size bin from the counts by the Condensation Particle Counter (Kangasluoma and Kontkanen, 2017). The inventory variables considered here are observed by combining several size bins observed by the DMPS. Inventory variable uncertainty is the uncertainty of the size distribution's corresponding size bins added in quadrature, which leads to inventory variables that are not as noisy as the
individually measured particle sizes. Since the inventory variables are defined as total concentration across a size range, the method intrinsically dampens instrument noise as random errors across multiple channels tend to cancel each other out.

The normalized standard deviation in the noisy $N_{3-6}, N_{10}, V_{dry}$ (with respect to their average value with noise) across the 27 scenarios is less than 0.08, 0.03, and 0.04 respectively. Then, we filter the synthetic noisy measurements with an 11 hr window and a first degree polynomial with the Savitzky Golay filter to produce measurement values and derivatives that are
smooth. As shown in Figure 6, the relative standard deviation of each inventory variable decreases to less than 0.03, 0.011, and 0.011 for $N_{3-6}, N_{10}, V_{dry}$ respectively. The filtered noisy synthetic measurements and their derivatives were used to estimate particle formation, emissions, and SOA production in the same 27 scenarios that were investigated in the previous section. Since we observe poor performance in 4 ill-conditioned scenarios, we focus on the remaining 23 well-conditioned scenarios in this section. The same version of TOMAS that was used above is used here, and the same estimation algorithm is
used except for additional code to handle the scenario when the system of equations is ill-conditioned. At each time step, we evaluate the condition number of the system of equations as in Equation (3) within the estimation algorithm; if its value is greater than three, the equation and unknown scaling factor corresponding to the row with the largest eigenvalue are removed. This corresponds to solving for just two uncertain process rates instead of three. When a scaling factor is removed from the system of equations, it then is assigned its value from the previous time step.

Figure 7 shows the mean bias and variance in the estimated process rates for each of the 23 scenarios in blue crosses and red circles when synthetic measurements without and with noise are used, respectively. In Figure 7a, we find that the normalized mean bias across the 23 scenarios does not significantly change, while the maximum bias in emissions among the 23 scenarios increases by an order of magnitude when noisy measurements are used. Figure 7b shows a statistical significance





in the variance of estimated SOA production and POA emission rates between the cases using measurements with and without noise. The estimated process rates using noisy measurements have a somewhat higher variance compared to the estimates with perfect measurements. The high variance in estimated process rates is due to the estimator tracking synthetic measurement noise, which is translated to noise in the process rates. In the future, the gain should be adjusted to a lower value so the

measurement noise is filtered and the estimated process rates are smoother.

## 4.    Estimation of Ambient Aerosol Dynamics

This section evaluates the inverse method by utilizing field measurements of particle size distribution from SPC, MPZ, and HYY instead of synthetic measurements to estimate particle formation, emissions, and SOA production. Since we previously found that an ill-conditioned sensitivity matrix results in inaccurate estimated process rates when using synthetic

measurements, we avoid solving an ill-conditioned system by reducing the system of equations. If the condition number of $G_k$ is greater than a threshold of five, then we assume the scaling factor for SOA production is constant from the previous time step, and estimate the other two scaling factors based on $N_{3-6}$ and $N_{10}$ inventory variables. Here, we eliminate the equation for $V_{dry}$ and eliminate the unknown scaling factor on SOA production because we observed that the full system of equations is ill-conditioned because $N_{10}$ and $V_{dry}$ are very close to co-linear. Since $N_{10}$ is more directly measured than $V_{dry}$, we choose to

remove $V_{dry}$.

The purpose of this section is to test the inverse method on real data, including size distributions not generated by the TOMAS model itself (the "synthetic measurements" above). A challenge here arises from the processes that are not well captured in a box modelling framework, namely long-range transport of aerosol to the measurement site, including abrupt changes in air mass. Recognizing that our long-term goal is to deploy the estimation framework in a three-dimensional model

that will include improved and more detailed representation of long-range transport, we mitigate them here with several simple approaches.

First, we filter the measurements to select time periods when meteorology is relatively stable. We classify whether a time is stagnant from the three conditions determined by Garrido-Perez et al. (2018) in addition to a condition on the sea level pressure. A time period is considered stagnant if 1) the reanalysis wind speed at 10 m altitude is less than 3.2 m s⁻¹, 2) the

reanalysis wind speed at 500 hPa is less than 13 m s⁻¹, 3) daily precipitation is less than 1 mm, and 4) sea level pressure is greater than 1020 hPa. The reanalysis precipitation fields are retrieved from the European Climate Assessment & Dataset E-OBS at a resolution of 0.25° latitude by 0.25° longitude (Cornes et al., 2018) while wind speed and sea level pressure are from the National Aeronautics and Space Administration's MERRA2 at 0.5° latitude by 0.625° longitude. All of the conditions must be met for at least 24 hours before the remaining hours fulfilling the constraints are considered stable. These conditions leave

us with 23, 31, and 18 days out of the one year of measurements at SPC, MPZ, and HYY, respectively. Out of the filtered measurements at each site, 24%, 43%, and 41% are from the spring season in SPC, MPZ, and HYY, respectively.





Second, we choose a first-order removal timescale that is faster than aerosol removal processes (i.e., deposition) to allow the box model to adjust to air mass changes. Finally, we use this largely as a proof-of-concept, taking caution in interpreting the process rates. We expect that, in a box modelling framework, the $N_{3-6}$ inventory variable and nucleation rates will be the most realistic of the process variables since nucleation mode particles are short-lived, and long-range transport is a relatively

minor factor in determining their concentrations. In contrast, we do not seek to interpret the emissions or SOA production rates since the inverse model will naturally adjust them artificially to compensate for transport processes not represented in the box model. Therefore for this work, we further simplify the box model as described in the next section.

### 4.1. Simulating Ambient Aerosol with TOMAS

The inverse modelling method assumes that all simulated processes in the box model are correct except for the processes scaled

by the control variables. Thus, the primary emitted size distribution must have the correct shape in order to estimate emissions correctly. The aerosol size distribution we emit into the TOMAS box model in this section reflects primary organic aerosol emissions from a 3D CTM (GEOS-Chem). Here, the particle emissions are from fossil fuel and biomass burning emission inventories, averaged over Western Europe (Bond et al., 2007).

Additionally, we remove condensation of sulphuric acid from the box model simulation so that we are estimating overall

particle growth while perturbing SOA production with the growth scaling factor. Using only this box model and measurements of particle size distribution, we cannot distinguish between sulphate and VOC condensation since both similarly affect the size distribution in the model. Since we have removed sulphuric acid from the box model, the default nucleation parameterization will not produce new particles. Instead, we replace the nucleation scheme with a constant new particle formation rate of 0.2 $cm^{-3}s^{-1}$. This will only affect the estimated scaling factors, but not the estimated nucleation rates.

### 4.2. Estimation Results

At each measurement site, the estimated inventory variables are close to the hourly measurement of those same parameters, as shown in Figure 8. At SPC, MPZ, and HYY the normalized absolute errors across the three inventory variables are 0.10, 0.14, and 0.13, respectively. Each horizontal panel in Figure 8 shows frequency scatter plots of the hourly estimated versus observed inventory variable where the color represents the count of data points in that respective grid cell.

Figure 9 shows the average estimated, measured, and original model predicted size distributions at each measurement site. Although the inventory variables are close to the observations, the estimated size distributions do not match as well as we found with the synthetic measurements. We expect that the estimated size distribution does not match the field measurements because mass and number of particles larger than 100 nm are long-range species that are influenced by processes not included in the version of TOMAS used here, i.e., various primary aerosol emission sources, transport, aerosol ageing. Several factors

contribute to the bias between the estimated and observed size distribution between 4 nm and 20 nm. First, TOMAS nucleates 3 nm particles based on the $N_{3-6}$ measurement in order to match the total number concentration in the 3 to 6 nm range to the observations, but the shape of the size distribution within that range does not match since all new particles enter through the 3





nm size bin. Second, the estimated SOA production rates exhibit similar high variability as the estimated nucleation rate in order to account for changes in air mass and wind direction not included in the model. Since peaks in the estimated growth from SOA production does not always coincide with nucleation events, the model simulation forms the two distinct modes at 3 nm and 100 nm, as shown in Figure 9.

We find that the estimated nucleation rates are reasonable as shown in Figure 10, which shows the average diurnal profile estimated for each site. The average estimated nucleation rate at all of the sites has a realistic magnitude near $1\ cm^{-3}s^{-1}$, which agrees with previous simulation results at HYY (Kulmala et al., 2005; Westervelt et al., 2014). MPZ and HYY have a clear peak in particle formation near noon local time, correlating with photochemical production of condensable vapours. At HYY we see an increase in estimated particle formation rate at 18:00 local time, which occurs on two dates: (February 22 and

March 28) once during daytime and once during night time. The estimated diurnal profile of formation at SPC includes significant formation even during night time. To understand the large, fairly consistent nucleation rate at SPC, we can examine the average diurnal profile of the measured $N_{3-6}$, shown in Figure SI.2. The fairly constant $N_{3-6}$ measurement throughout the day may indicate an instrument bias, which introduces a similar bias into the estimated nucleation rate at SPC.

**Conclusions**

This work has explored a way to integrate particle size distribution data with an aerosol microphysics algorithm used in 3D CTMs by designing a novel estimation algorithm borrowed from the field of nonlinear process control. The estimation framework is robust, computationally inexpensive, and flexible to model updates. It has been tested with synthetic measurements, noisy synthetic measurements, and European field measurements. We show that the particle size distribution inverse modelling technique estimates particle formation, emissions, and SOA production accurately when there is no

measurement error and all other processes are known accurately. $N_{3-6}$, $N_{10}$, and $V_{dry}$ are estimated within 11% error of the synthetic measurements across 24 realistic scenarios that span global conditions. New particle formation, emissions, and growth rates were estimated within 13% of the true rates in 23 scenarios that are well-conditioned. Moreover, the estimated size distribution matches the synthetic measurements when the inventory variables are accurately estimated. Introducing realistic instrumental noise into the synthetic measurements results in a statistically significant increase in the variance of the

estimated SOA production and emission rates. Nonetheless, no significant biases were introduced in the estimated process rates across the 23 well-conditioned scenarios when adding noise to the synthetic measurements.

We applied the inverse technique to field data from San Pietro Capofiume, Melpitz, and Hyytiälä between May 2006 and June 2007. Error in the estimated $N_{3-6}$, $N_{10}$, and $V_{dry}$ is less than 15% during stagnant measurement days in which the meteorology is most amenable to a box modelling framework. Although the estimated emissions and growth are not necessarily

accurate since the box model cannot represent long-range transport of aerosol resulting from these processes, we can estimate the nucleation rate. The average estimated nucleation rates at Melpitz and Hyytiälä have peaks near noon local time with



average values of 1 and 0.15 $\text{cm}^{-3}\text{s}^{-1}$, respectively. This application demonstrates the ability of our method to derive reasonable process rates from long-term aerosol size distribution measurements.

Although there is reason to believe that the estimated emissions and SOA production rates are not correct when applying the box model to field measurements, these rates are estimated correctly in the synthetic measurement case. The key differences

between field and synthetic measurements are the model-measurement biases outside of particle formation, emissions, and growth. The zero-dimensional version of TOMAS used here does not incorporate sufficient detail about meteorology or emission sources to give successful inverse modelling results. However, a 3D CTM that includes processes such as transport, photochemistry, diverse emission sources, etc., could be more successful in an inverse modelling study. If the method applied here is integrated into a 3D CTM, there is potential to estimate key uncertain parameters that control aerosol dynamics and,

thereby, improve the predicted size distribution field.

## Appendix A: Uncertainty of Particle Size Distribution

The DMPS takes aerosol sample and reports particle count of a specific size bin during a time interval, as represented by the following equation:

$$\mathbf{c} = \mathbf{M} \cdot \mathbf{n} \tag{A-1}$$

where $\mathbf{c} = (c_1, c_2, \cdots c_I)^\mathrm{T}$ represents the counts of particles in each size bin, $\mathbf{n} = (n_1, n_2, \cdots n_J)^\mathrm{T}$ is the actual particle size distribution, and $\mathbf{M}$ is the matrix involving all the processes inside the DMPS (Pfeifer et al., 2014): the charging probability, the transfer function, and the counting efficiency of the detector. The charging probability follows the empirical approximation by Wiedensohler (1988). For a specific voltage, flow rates, and the geometry of the differential mobility analyzer (DMA), the DMPS system is at steady state, so the Stolzerburg (1988) transfer function can be used to approximate the transmission

efficiency of particles at the corresponding conditions. The counting efficiency of the detector is determined by the parameters from Mertes (1995).

We assume the uncertainty of the measurement only comes from the counting, which follows Poisson counting statistics, since the information about the uncertainty of the working conditions is unknown. Thus, the uncertainty from measurement is $\sqrt{\mathbf{c}}$, which yields the uncertainty of the particle size distribution, $\mathbf{E_n}$, as:

$$\mathbf{E_n} = \mathbf{M}^{-1} \cdot \sqrt{\boldsymbol{c}} \tag{A-2}$$

The model generates the matrix $\mathbf{M}$, and with the input of the actual particle size distribution $\mathbf{n}$, first calculates the counts $\mathbf{c}$ and its element-wise square root ($\sqrt{\mathbf{c}}$ ), and then applies the totally nonnegative least squares method (Merritt and Zhang, 2005) to obtain the uncertainty of the particle size distribution $\mathbf{E_n}$.



**Author Contributions**

PJA, BEY, and DLM conceptualized the simulations and estimation technique. DLM adapted the model code with the estimation algorithm and performed all simulations. RF and YH conceptualized the measurement uncertainty. YH created the uncertainty code. DLM authored the text with contributions from all co-authors. YH wrote Appendix A.

**Code and Data Availability**

The code used in this work is available at https://github.com/danamcguff/TOMAS-InverseModel. Data on particle size distribution measurements were provided through EBAS (http://ebas.nilu.no) for the MPZ and HYY stations. Stefano Decesari provided size distribution measurements for the SPC station. Precipitation data was downloaded from European Climate Assessment & Dataset (https://ecad.eu/download/ensembles/download.php). Sea level pressure and wind speed data was

downloaded from NASA Goddard Earth Sciences Data and Information Center (https://disc.gsfc.nasa.gov).

**Acknowledgements**

The authors acknowledge Thomas Tuch, Pasi Aalto, Stefano Decesari, and Jorma Joutsensaari for providing descriptions and technical specifications of their measurement set-up. This work was performed partially under the auspices of the U.S. Department of Energy by Lawrence Livermore National Laboratory under Contract DE-AC52-07NA27344 with IM release

number LLNL-JRNL-813683-DRAFT.

**Financial support**

This research was supported by the Carnegie Mellon University department of Chemical Engineering and the Mahmood I. Bhutta Fellowship in Chemical Engineering.

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



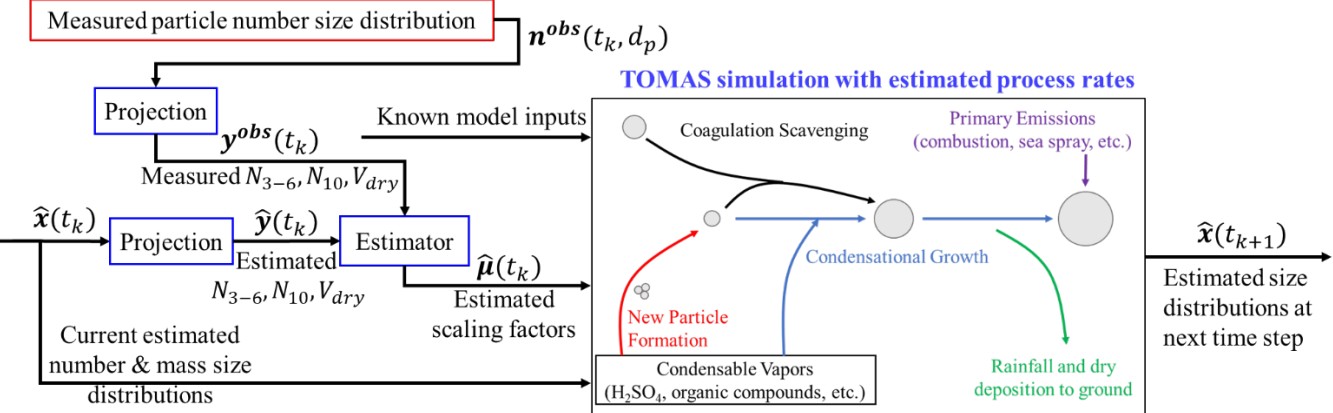

**Figure 1. Schematic of TOMAS box model with online estimation technique. The measured size distribution is projected to the inventory variables ($N_{3-6}, N_{10}, V_{dry}$), which the estimator compares to the current model prediction to calculate scaling factors that adjust particle formation, SOA production, and POA emission rates in TOMAS to simulate the size distribution with adjusted rates at the next time step.**

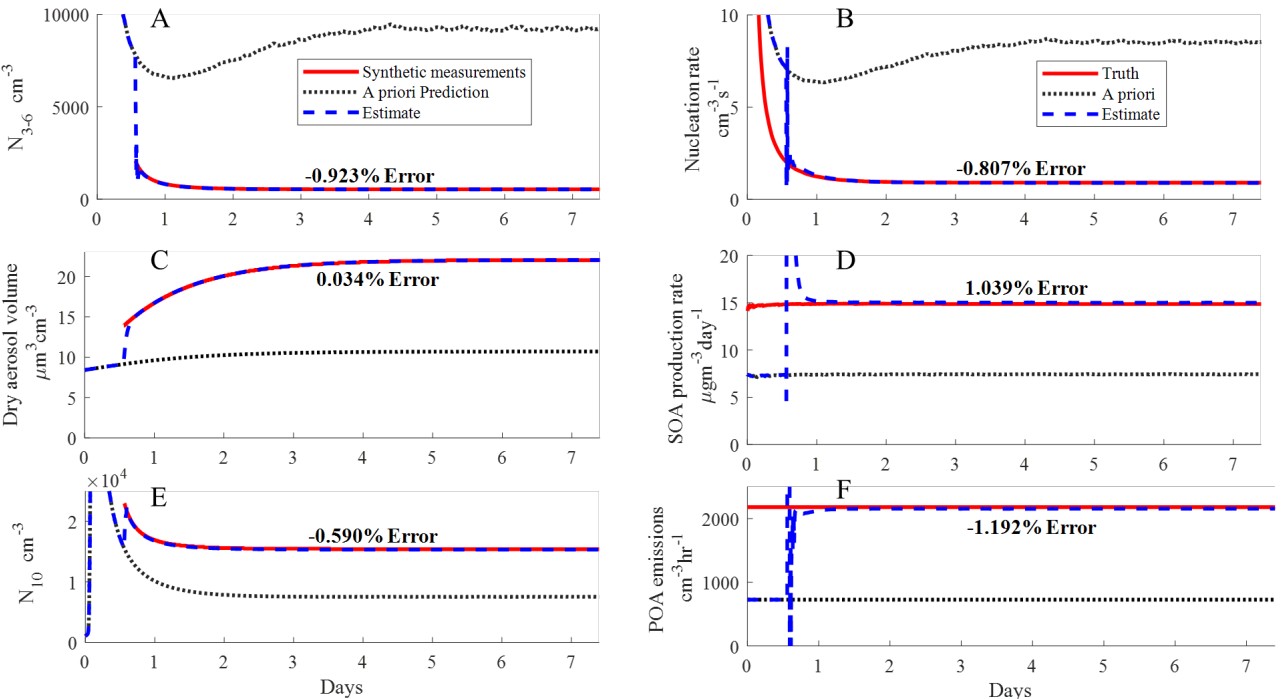

**Figure 2. Estimated, measured, and *a priori* inventory variables in left panel and estimated, true, and *a priori* process rights in right panel for scenario #26 of proof-of-concept scenarios. The measurements are (a) $N_{3-6}$, (c) $V_{dry}$, and (e) $N_{10}$. The process rates are (b) nucleation rate, (d) SOA production rate, and (f) POA emission rate.**

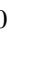
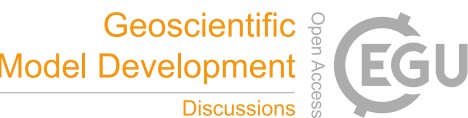

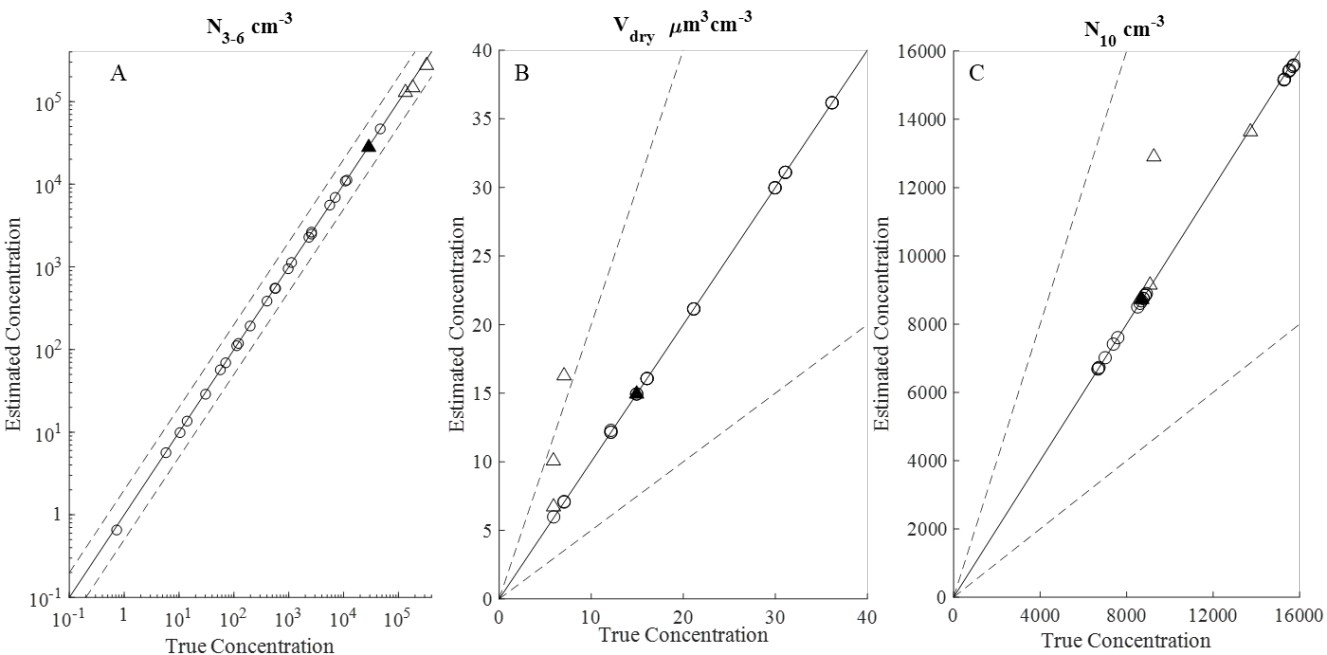

**Figure 3. Scatter plots of time-averaged (a) N₃₋₆, (b) V_dry, and (c) N₁₀ comparing the inverse modelling estimate and truth from synthetic measurements for each scenario. Circle and triangle markers represent well- and ill-conditioned scenarios, respectively. Open triangles represent scenarios unlikely to find with field measurements. Solid line is 1:1 and dashed lines are 1:2 and 2:1.**

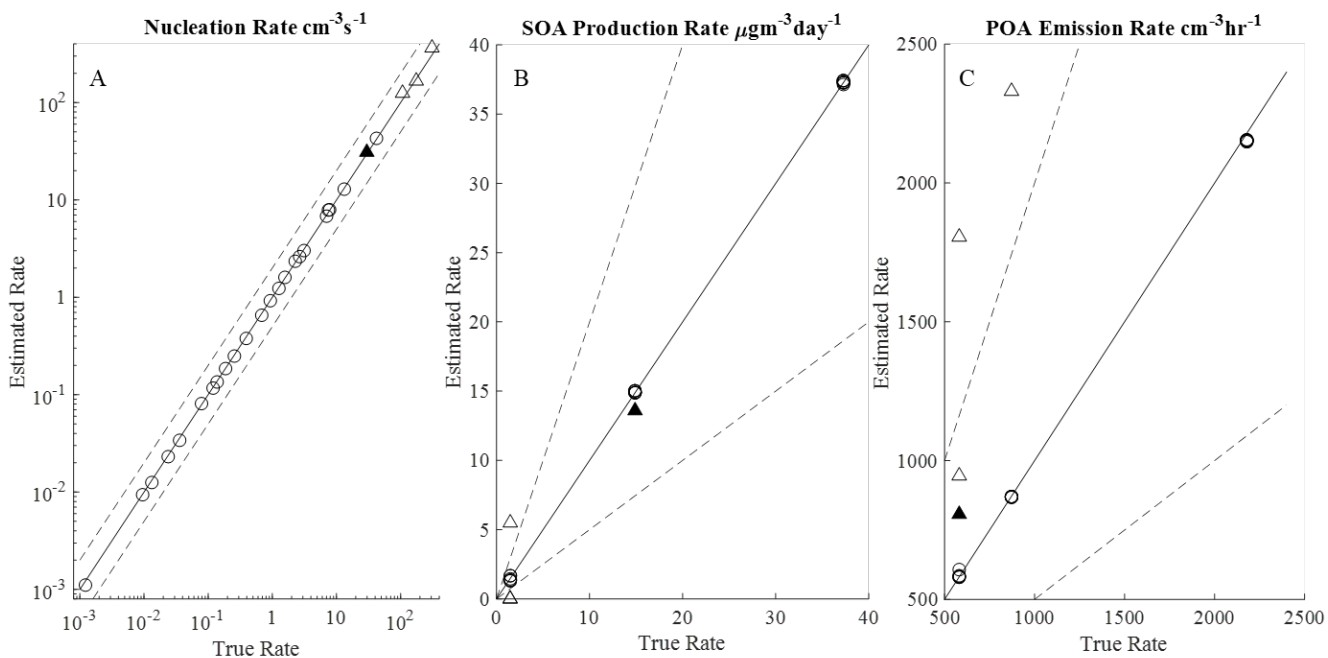

**Figure 4. Scatter plots of time-averaged nucleation rate (a), SOA production rate (b), and POA emission rate (c) comparing PBIO estimate and truth for each scenario. See Figure 3 for details.**



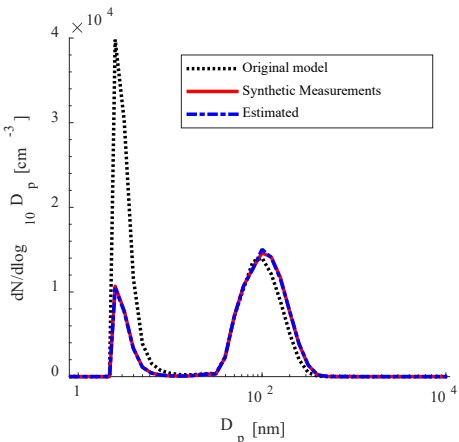

**Figure 5. Average particle size distribution in scenario #14 (Nucleation, SOA production, POA emission rates: 2.7 cm$^{-3}$s$^{-1}$, 15 μg m$^{-3}$day$^{-1}$, 872 cm$^{-3}$hr$^{-1}$) from synthetic measurements, TOMAS run with *a priori* rates, and estimated with the inverse model are shown in the solid red, dotted black, and dashed blue profiles, respectively.**

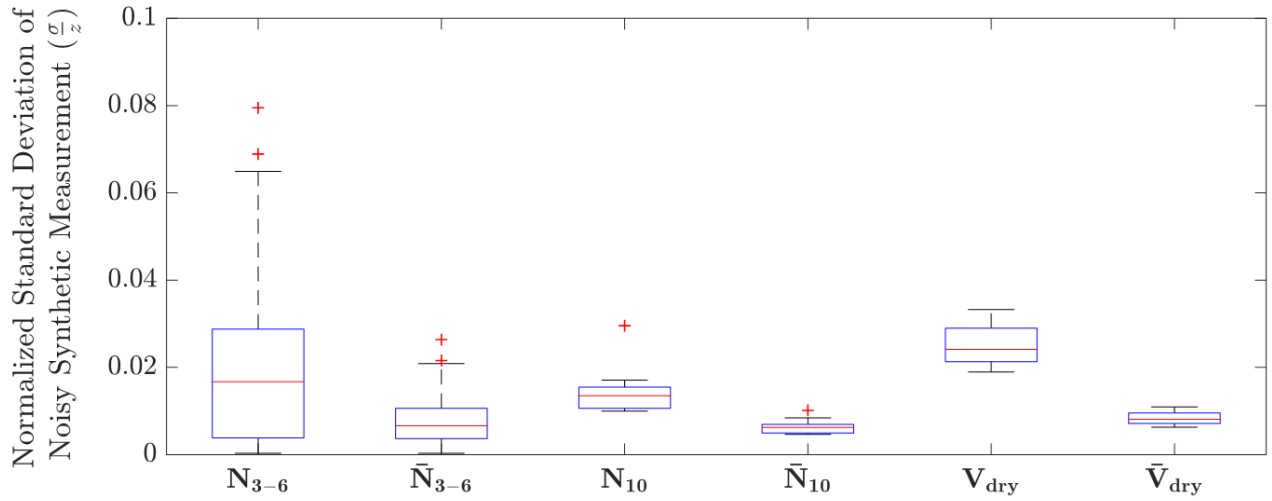

**Figure 6. Box plot showing standard deviation of "noisy synthetic measurements" relative to their average value in all 27 scenarios for the inventory variables originally ($N_{3-6}$, $N_{10}$, $V_{dry}$) and after filtering ($\bar{N}_{3-6}$, $\bar{N}_{10}$, $\bar{V}_{dry}$).**





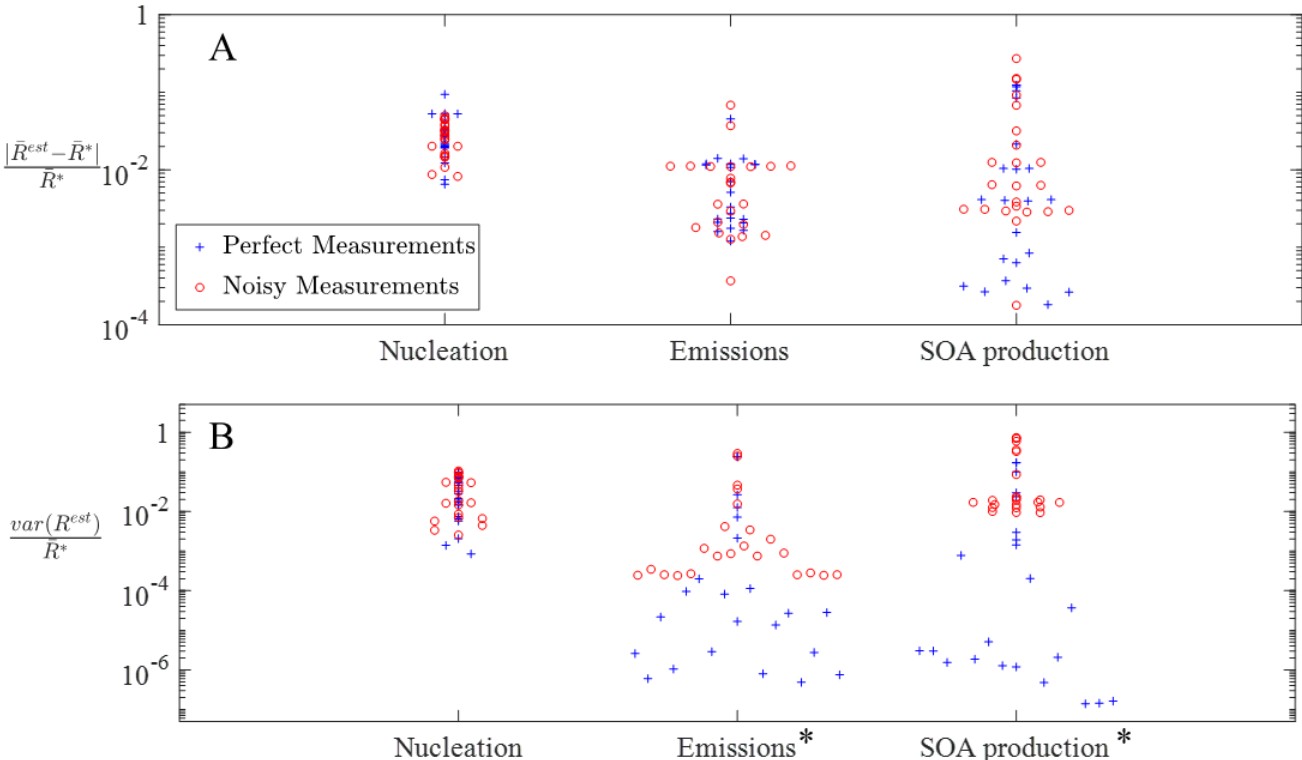

**Figure 7. Beeswarm plots showing (a) normalized mean error and (b) normalized variance in estimated nucleation, emissions, and SOA production rates ($R^{est}$) relative to the actual process rates ($R^*$). The blue crosses show the 23 scenarios with perfect measurements and the red circles show the same 23 scenarios with noisy measurements. *Difference between the perfect and noisy measurement scenarios is statistically significant for $\alpha = 0.05$.**



**Figure 8. Frequency scatter plots comparing hourly estimates to hourly measurements of N$_{3-6}$, N$_{10}$, and V$_{dry}$ in the first, second, and third columns, respectively. Measurements from San Pietro Capofiume, Melpitz, and Hyytiälä are shown in the first, second, and third rows, respectively. Black line show 1:1 line. Normalized absolute error (NAE) is shown at top of each figure.**



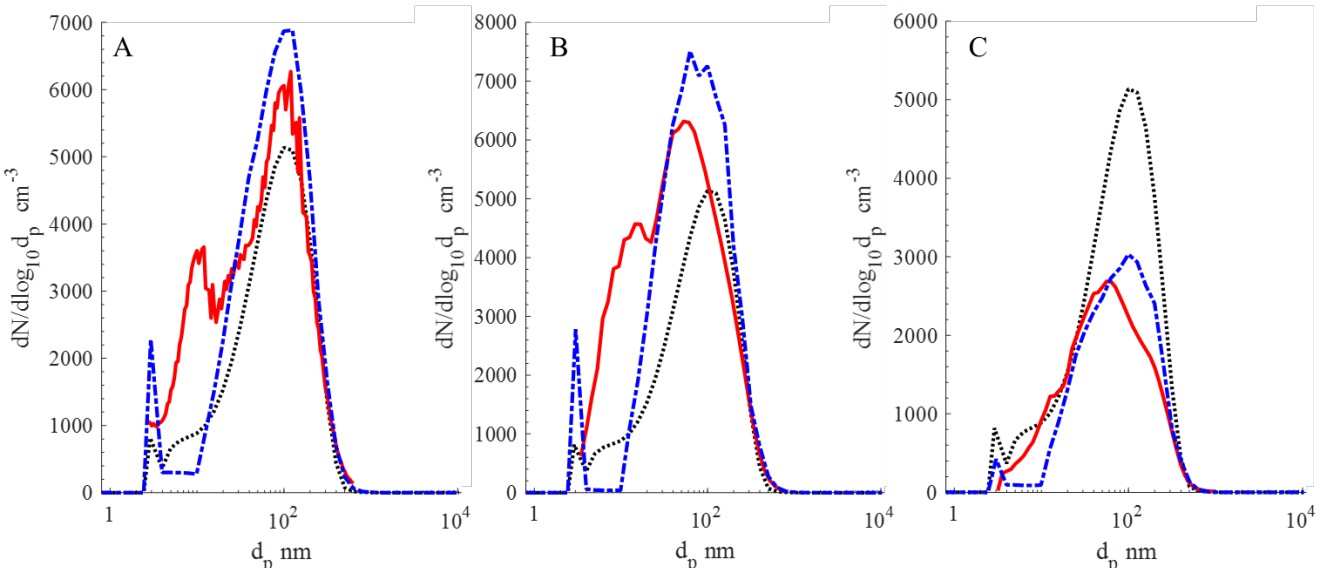

**Figure 9. Average particle size distributions measured, originally predicted with *a priori* rates in TOMAS, and estimated with the inverse model are shown in the solid red, dotted black, and dashed blue profiles, respectively. Results are shown using measurements at (a) San Pietro Capofiume, (b) Melpitz, and (c) Hyytiälä.**

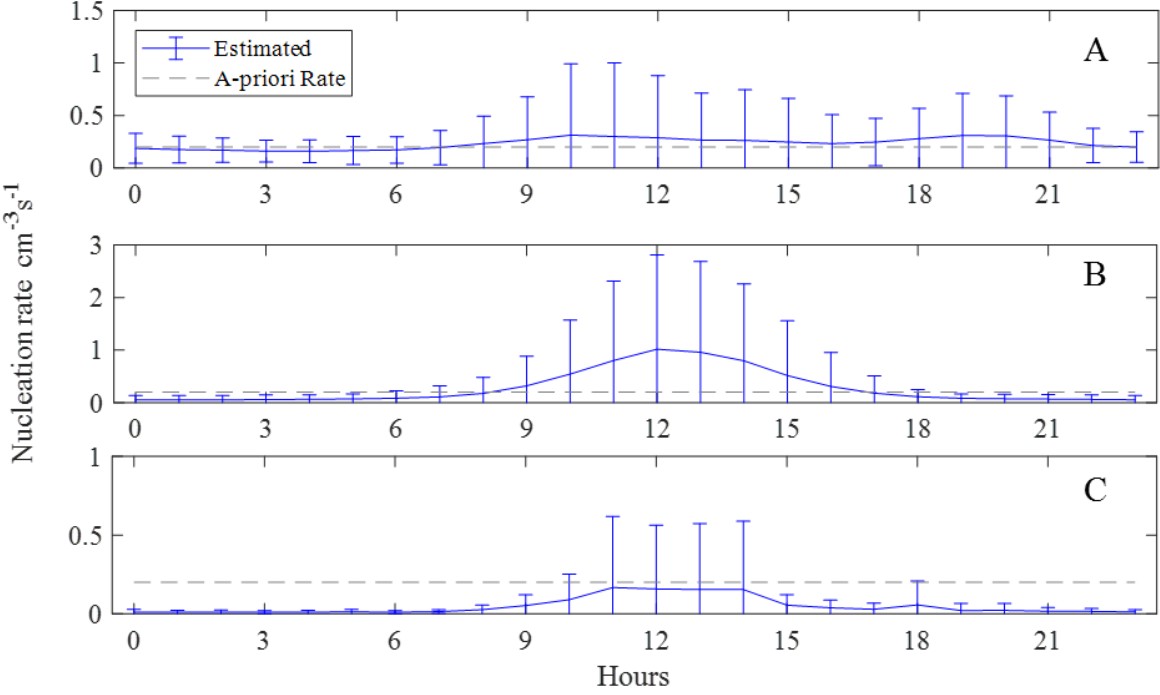

**Figure 10. Estimated diurnal nucleation rate during stagnant events for each measurement site: (a) San Pietro Capofiume, (b) Melpitz, (c) Hyytiälä. Blue solid and grey dashed lines show the estimated rate and *a priori* rate, respectively, the error bars show standard deviation across all days.**