# Peer review of "Novel estimation of aerosol processes with particle size distribution measurements: a case study with TOMAS algorithm v1.0.0"

_Geoscientific Model Development, 2020_

## Short Comment (SC1) · 10 Sep 2020

Dear authors,

in my role as Executive editor of GMD, I would like to bring to your attention our Editorial version 1.2:

https://www.geosci-model-dev.net/12/2215/2019/

This highlights some requirements of papers published in GMD, which is also available on the GMD website in the 'Manuscript Types' section:

http://www.geoscientific-model-development.net/submission/manuscript_types.html

[Figure]

In particular, please note that for your paper, the following requirements have not been met in the Discussions paper:

- "The main paper must give the model name and version number (or other unique identifier) in the title."

- "Code must be published on a persistent public archive with a unique identifier for the exact model version described in the paper or uploaded to the supplement, unless this is impossible for reasons beyond the control of authors. All papers must include a section, at the end of the paper, entitled "Code availability". Here, either instructions for obtaining the code, or the reasons why the code is not available should be clearly stated. It is preferred for the code to be uploaded as a supplement or to be made available at a data repository with an associated DOI (digital object identifier) for the exact model version described in the paper. Alternatively, for established models, there may be an existing means of accessing the code through a particular system. In this case, there must exist a means of permanently accessing the precise model version described in the paper. In some cases, authors may prefer to put models on their own website, or to act as a point of contact for obtaining the code. Given the impermanence of websites and email addresses, this is not encouraged, and authors should consider improving the availability with a more permanent arrangement. Making code available through personal websites or via email contact to the authors is not sufficient. After the paper is accepted the model archive should be updated to include a link to the GMD paper."

Thus please provide the version number of the TOMAS-algorithm in the title of your article upon revision and ensure to permanently archive the exact Code version used for this publication. Note, that github is not a permanent archive. As explained in https://www.geoscientific-model-development.net/about/manuscript_types.html the preferred reference to this release is through the use of a DOI which then can be cited

in the paper. For projects in GitHub a DOI for a released code version can easily be created using Zenodo, see https://guides.github.com/activities/citable-code/ for details.

Yours,

Astrid Kerkweg

---

## Referee Comment (RC1) · Anonymous Referee #1 · 10 Oct 2020

Review of gmd-2020-281: "Novel estimation of aerosol processes with particle size distribution measurements: a case study with TOMAS algorithm"

General comments

In this work the authors present and examine a proposed new method for incorporation of information from aerosol particle size distribution measurements into atmospheric chemical transport models (CTMs). Specifically, the method utilizes an inverse modeling algorithm to constrain and adjust estimates of relevant aerosol microphysical process rates based on changes to the size distribution and the use of the TOMAS aerosol microphysics model. Aerosol nucleation, emissions, and growth processes are

investigated here using a simple "box model" to assess the utility and efficiency of the method. The stated goal of the work is to investigate if such a method could potentially be utilized in a 3D CTM to better constrain estimates of such aerosol process rates.

The manuscript is generally well written and understandable, and includes reasonable first steps to investigate the utility of the method. The authors select three "inventory variables", which can be derived directly from measured size distributions, and use the inverse model to link changes to these variables with changes to the various aerosol process rates. In using synthetic measurements (including reasonable estimates for measurement noise), they demonstrate that under conditions in which only the investigated aerosol processes are relevant to changing size distributions, the method can successfully estimate the various underlying process rates. When applying the method to actual observed measurements the estimates of process rates expected to be sensitive to additional atmospheric and aerosol processes are less successful, though this is expected for the simple box model scenario. The result is a successful demonstration of the potential of the method, though it remains to be seen (in future work) if information from actual size distribution measurements can be incorporated into a 3D CTM where more degrees of freedom may make inverse modeling more difficult.

Extraction of information from aerosol particle size distributions can be particularly difficult as many underlying processes—not all of which are well measured in the actual atmosphere or well represented in idealized CTMs—contribute to changes in both the aerosol particle size distribution and the derived inventory variables used here. As such, significant caution is warranted when assessing efforts to do so. However, efforts to include better estimates of more fundamental aerosol process (such as the nucleation, emission, and growth rates investigated in this work), as opposed to variables that serve more as proxies for actual aerosol processes (such as aerosol species mixing ratios, as noted by the authors), are valuable contributions to aerosol modeling efforts. In this regard, the authors may in fact be underselling the potential of their results to some extent, as computationally efficient methods to better incorporate such

fundamental aerosol processes into 3D CTMs is both important and valuable.

As such, I recommend the manuscript be published in Geoscientific Model Development subject to minor corrections and consideration of several questions I list below.

Specific comments

Several related open questions remained that appear to not be fully addressed within this work. While they are likely beyond the scope of this manuscript and need not be fully resolved here, it may be worthwhile for the authors to include a discussion of them for efforts moving forward with this method.

1. The question of the relevance of other aerosol processes on process rate estimates from the inverse model.

In particular, a set of measured aerosol particle size distributions in the actual atmosphere will be subject to variation from a wide variety of processes. Some of these can be simulated by either the CTM or TOMAS to varying degrees of accuracy while others may not be included at all. As a first step the authors select three aerosol processes for their relevance to prediction of CCN in a CTM (page 2-7). In circumstances where other processes are relevant to changing aerosol particle size distributions—perhaps terms included in f_k, or other processes held constant in the model but changing in the actual atmosphere and therefore affecting the measured distribution—the result would presumably be a (not physically relevant) change to at least one of these investigated process rates. Is there a limit to how much the resulting error on the estimated process rate would be, or a method to identify when such a circumstance is occurring? Would it be evident in, say, the condition number of equation 3, or is there another method of detecting such a scenario and limiting its impact on process rate estimates?

This is of course an issue with all under-constrained inverse problems, not unique to this method, and is beyond the scope of this manuscript to resolve. However, some discussion of the issue may be warranted as an outstanding issue with such efforts.

Interactive
comment

2. Additional physical reasoning regarding the circumstances under which an ill-conditioned sensitivity matrix may occur.

The authors discuss a weakness of the method occurring when the sensitivity matrix, $G_k$, is ill conditioned (page 9-6). Cases in which solutions to these situations are discussed occurs at page 10-28 and 11-10. In these situations, it may be helpful to include more physical reasoning regarding the ill-conditioned scenario and the implications of removing one uncertain process rate (held constant for the given time step).

To a very general degree, additional discussion might address several open questions: Would this be interpreted as the aerosol process being generally sensitive to changes in more than one inventory variable? What are the physical implications of this in terms of adjustments to the aerosol process rates at such time steps? Would real world changes in other relevant processes, or those contained in $f_k$, be at least partially responsible? Moving forward with this method, should those be considered? How does this model perform when multiple process rates are changing simultaneously?

For example, in the ill-conditioned scenario described at the beginning of section 4, would the physical interpretation be that changes to both emissions and SOA production rates are expected by TOMAS to have roughly the same impact on $N_{10}$ and $V_{dry}$ (similar sensitivities in $G_k$)—and that information on changes to the size distribution (via our inventory variables) are constrained to only influence emission rate to solve this issue? That would seem to be a reasonable solution when lacking a better reason to constrain one of the process rates, but more discussion of $G_k$ and the condition number in such circumstances would be helpful. If this is not the correct interpretation of these ill-conditioned sensitivity scenarios, it may help to provide more physical reasoning of what is happening in the model and what the physical implications are in these circumstances.

Minor comments and technical corrections

1. When referring to the three investigated aerosol processes (nucleation, emis-

sions, and growth) throughout the manuscript, the order of these terms is occasionally changed (e.g. abstract line 1-14 "... aerosol nucleation, emissions, and growth rates..." and 1-19 "... aerosol emissions, growth, and nucleation..."). Readability may be improved simply by keeping a consistent order when referring to these processes.

2-14: The last several sentences in this paragraph are somewhat confusingly worded. Consider clarifying the meaning here.

3-1: The use of the word "integrates" could be misconstrued as integration of the size distribution here. Consider a different word such as combines or includes.

4-14: "as nearly as possible". Consider something like "as closely as possible".

4-19: The point of this sentence is valid, but these processes are not the only processes that control evolution of the particle size distribution.

6-2: The terms here look somewhat like t_k(mu_k) as a function. Consider a change to something like "...depend on the scaling factors (mu_k), at time t_k ..." if appropriate.

7-1: Should mu_k_hat be referred to as "scaling factors" or similar here, as in 6-25 and 7-11, to prevent confusion?

7-17: Even if restating from an earlier cited source, an additional reference for more information on the use of the condition number and relative sensitivity array in this methodology would be helpful for a reader wanting more information on this step.

7-24: Consider "e.g." rather than "i.e." if appropriate.

8-25: Refer to inventory variables for each of these in this sentence, i.e. "...under-predicted aerosol mass (via dry aerosol volume, V_dry) as well as N_10..." or similar.

9-5: What was the justification for these ranges of rates spanning expected atmospherically relevant process rates?

Section 3.1: Was the uncertainty and estimated effect of instrument noise calculated

using only the limited "meteorologically stable" time periods as opposed to the full year of data?

10-16: Should be "damps" rather than "dampens".

13-15: Use of "integrate" again here. Consider "... way to combine information from ..." or similar instead.

Fig 9: Needs a legend to show line color meaning in figure.

---

## Referee Comment (RC2) · Anonymous Referee #2 · 6 Nov 2020

In this work, the authors apply an inverse modelling method based on inventory control to an aerosol box model. The application of this method to aerosol nucleation and growth is novel and interesting, and I look forward to seeing what insights can be gained from its application in a more comprehensive modelling study. The paper is well-written, and merits publication provided that my following comments are addressed:

General comments:

In reality, emissions (and other processes) vary on a wide range of time scales, including seasonal, weekly, and diurnal cycles, and variability at shorter timescales. These variations are likely not fully captured by any CTM. Does this present a challenge for

the choice of an appropriate value for the gain $K_c$? Can the authors comment on the robustness of their method to this issue?

Could the authors discuss the sensitivity of the results to the sink rate chosen in this work? I understand that a detailed treatment of dilution, transport, and deposition is beyond the scope of this study. However, given that the rates of these processes are both variable and uncertain, I think that a little further discussion is warranted. This discussion would also inform the potential of this method to be applied in a CTM, as there are uncertainties in more model processes than can be tested simultaneously using this method.

Technical comments:

I do not see the chosen values of the gain $K_c$ listed in the paper. It would be best to list them in section 2.2. It may be helpful to express them as convergence timescales.

p4, line 20: Is it improved performance that the authors anticipate, or greater understanding? I would guess that the authors would find similar or better performance in using the mixing ratios directly as the control variables.

p8, line 10: What is the timescale of the moving average? The authors later state that the synthetic noisy measurements are filtered with an 11-hour timescale. Was the same timescale applied to the observations?

p9, line 1, "will repeat": The authors should use the present tense here. Reserve the future tense for future work.

p10, line 31-33. I found this sentence confusing. If I am reading Fig. 7a correctly, the maximum in normalised mean bias increases from 0.06 to 0.09. Is the "maximum bias" the authors are referring to, then, the bias for a single time step of the box model (not shown in the figures)? If so, I would request that this be split into two sentences, as the second half does not refer to Fig. 7a.

p5, line 1 states that the TOMAS model simulates particles as small as 0.5 nm in diameter. However, p12, line 31 seems to indicate that nucleated particles are generated with an initial diameter of 3 nm. Are the smallest size bins unused (i.e. always contain zero mass) in this study? This should be stated plainly in the methods section.

p13, lines 9-10: It may be clearer to say "this is after sunset on February 22 and before sunset on March 28th".

Figure 7: Does the mean bias in the nucleation rate decrease when noise is added to the synthetic measurements? If so, this warrants a brief discussion in the text.

---

## Referee Comment (RC3) · Anonymous Referee #3 · 25 Nov 2020

Review of "Novel estimation of aerosol processes with particle size distribution measurements: a case study with TOMAS algorithm"

The authors employ an inverse modeling technique on an aerosol microphysical model in order to scale uncertain simulated aerosol processes (nucleation, emissions, and growth) to improve simulated aerosol properties (N3-6, N10, Vdry) compared to observations. As an initial step, the authors test the inverse modeling technique on synthetic data with and without noise as well as observed aerosol size distributions in Europe. The approach is novel, interesting, and has potential towards broader applications (as noted in the manuscript). The manuscript is well written and the results are convincing. I believe the manuscript is suitable for publication after consideration of the minor comments outlined below.

General Comments:

I believe the manuscript would benefit from further discussion on the limitations of applying this approach to ambient size distributions which may be influenced by many uncertain aerosol processes that are not being scaled in the inverse technique. As aerosol processes are often non-linear, how sensitive is this method to potential errors in the representation of other aerosol processes? The authors do introduce this issue in Section 4.2 (and I agree a full exploration of the problem is beyond the scope of this paper). What are the implications of the assumption that the other modeled aerosol processes are correct? If a given aerosol process is drastically misrepresented in the CTM, will this inverse approach overcompensate (attempting to get the correct answer for the wrong reason)?

How generalizable is this approach in terms of choosing the scale factors and inventory variables? Would it be relatively straightforward for future studies to choose different aerosol processes to scale (for instance, if I wanted to assume nucleation rates are accurate but instead scale dry deposition rates)?

How is the exponential error decay factor (Kc) tuned? Is it kept constant across the simulations using the synthetic and observed data or is it tuned in each simulation?

Specific comments:

1. Line 2-13 could be rephrased as there are other processes that could contribute to aerosol growth not considered here.

2. What is the normalized error for the aerosol properties simulated with the a priori TOMAS model in Figure 8? How does this compare when using the inverse method?

3. I think Figure 9 could benefit from a legend or additional annotation. I found it hard to remember each color representation.

---

## Author Comment (AC1) · 14 Jan 2021

**Author Responses for GMD-2020-281.**

**SC1: 'executive editor comment on gmd-2020-281'**

We have named the algorithm used as version v1.0.0 to meet the journal requirements. This is now included in the title and in the Zenodo reference. The DOI is cited in the "Code Availability" section (15-24).

**Referee Comments**

Thank you all for your comments and feedback. Here are some specific responses and references to revisions in the text with the author responses in bold below each referee comment.

**RC1: 'Review of gmd-2020-281'**

"1. The question of the relevance of other aerosol processes on process rate estimates from the inverse model.
In particular, a set of measured aerosol particle size distributions in the actual atmosphere will be subject to variation from a wide variety of processes. Some of these can be simulated by either the CTM or TOMAS to varying degrees of accuracy while others may not be included at all. As a first step the authors select three aerosol processes for their relevance to prediction of CCN in a CTM (page 2-7). In circumstances where other processes are relevant to changing aerosol particle size distributions—perhaps terms included in f_k, or other processes held constant in the model but changing in the actual atmosphere and therefore affecting the measured distribution—the result would presumably be a (not physically relevant) change to at least one of these investigated process rates. Is there a limit to how much the resulting error on the estimated process rate would be, or a method to identify when such a circumstance is occurring? Would it be evident in, say, the condition number of equation 3, or is there another method of detecting such a scenario and limiting its impact on process rate estimates?
This is of course an issue with all under-constrained inverse problems, not unique to this method, and is beyond the scope of this manuscript to resolve. However, some discussion of the issue may be warranted as an outstanding issue with such efforts."

**The sensitivity of inventory variables to model errors determines the magnitude that the unknown model error affects the estimated rates. Thus, there is not a general rule of thumb on how much error this issue would contribute to the estimated rates, but it depends on the specific scenario (e.g., with an inventory variable of total aerosol volume: coagulation errors would not affect the estimates while wet deposition errors would strongly affect estimates). As you mentioned, the only way we can think of identifying such a situation is if the estimated rates are not physical. This takes some expert domain knowledge of the process and location being simulated. We have added a paragraph discussing this starting at 7-29, which states:**

*Another drawback of this estimation method, which is shared with most inverse techniques, is the effect of uncertainty in model errors not corrected by the estimator ($f_k$ uncertainty). Model errors outside of particle nucleation, emissions, and growth can lead the estimator to over-compensate in*

*the scaling factors if the inventory variables are sensitive to those model errors. For example, the estimation technique applied to a model that does not correctly simulate particle deposition will estimate particle emissions incorrectly while the estimated nucleation rate will not be affected. Identifying this scenario with outside model errors negatively impacting the estimated process rates is very difficult, but one sign is when the estimated process rates are not physical, e.g., a higher SOA production rate during nighttime. In future work, including various types of observations and more inventory variables can possibly inform the scaling factors and limit the impact of outside model errors on the estimated process rates.*

**The condition number would not necessarily provide insight on model errors since it is calculated by just the sensitivity array (G_k), which is only indirectly effected by model errors.**

> "2. Additional physical reasoning regarding the circumstances under which an ill-conditioned sensitivity matrix may occur.
> The authors discuss a weakness of the method occurring when the sensitivity matrix, G_k, is ill conditioned (page 9-6). Cases in which solutions to these situations are discussed occurs at page 10-28 and 11-10. In these situations, it may be helpful to include more physical reasoning regarding the ill-conditioned scenario and the implications of removing one uncertain process rate (held constant for the given time step)."

**We have added further clarification on the condition number and the physical meaning behind an ill-conditioned system at 7-16, which should clarify the questions raised below as well. This section now states:**
*This corresponds to the situation in which the measured inventory variables do not unambiguously constrain the process rates, i.e. several sets of process rates adequately satisfy the measured constraints, leaving at least two of the equations nearly linearly dependent. Physically, this can be thought of as a scenario in which inventory variables react to an aerosol process in the same way. For example, the system of equations using inventory variables $N_{10}$ and $N_{100}$ to estimate SOA production and emissions would be ill-conditioned in a scenario with the model predicting only particles greater than 100 nm and an emitted size distribution of particles 100 nm and larger. It follows that the condition number is not informed by any model-measurement mismatch.*

> "To a very general degree, additional discussion might address several open questions: Would this be interpreted as the aerosol process being generally sensitive to changes in more than one inventory variable? What are the physical implications of this in terms of adjustments to the aerosol process rates at such time steps?"

**The ill-conditioned case should be interpreted as: changes in aerosol processes affect more than one inventory variable in the same way. That is to say, as SOA production and POA emissions are increased, the increase in N10 due to each of those processes is the same or a factor times the increase in Vdry due to each of those processes. Since we are dealing with this by removing an aerosol process and inventory variable from the 3x3 system of equations, the implication is that we are only able to constrain two of the uncertain process rates while the remaining process evolves as it was in the previous timestep even if that is unrealistic.**

"Would real world changes in other relevant processes, or those contained in f_k, be at least partially responsible? Moving forward with this method, should those be considered?"

**No, the sensitivity and condition number is calculated solely based on the model equations, so any real world changes to other processes not captured in the model do not affect this calculation of the condition number.**

"How does this model perform when multiple process rates are changing simultaneously?"

**In each of the 27 scenarios with synthetic measurements, all three process rates are scaled simultaneously, so the estimation technique performs very well.**

"For example, in the ill-conditioned scenario described at the beginning of section 4, would the physical interpretation be that changes to both emissions and SOA production rates are expected by TOMAS to have roughly the same impact on N_10 and V_dry (similar sensitivities in G_k)—and that information on changes to the size distribution (via our inventory variables) are constrained to only influence emission rate to solve this issue? That would seem to be a reasonable solution when lacking a better reason to constrain one of the process rates, but more discussion of G_k and the condition number in such circumstances would be helpful. If this is not the correct interpretation of these ill-conditioned sensitivity scenarios, it may help to provide more physical reasoning of what is happening in the model and what the physical implications are in these circumstances."

**In this case, we still take into account changes to the inventory variables due to SOA production, but we do not have enough information (through the inventory variables) to determine both POA and SOA rates. So, here we assume the modeled SOA production is correct and assume all the model-measurement bias is due to POA to estimate POA emission rate.**

2-14: The last several sentences in this paragraph are somewhat confusingly worded. Consider clarifying the meaning here."
**The wording is slightly changed to be clearer at 2-14, which now reads:**

*An uncertain amount of VOCs are emitted from biomass burning, anthropogenic sources, and the biosphere (Folberth et al., 2006). After their emission, sulphuric acid and VOCs form secondary organic aerosol (SOA) (e.g. Kerminen et al., 2018; Kulmala et al., 2014; Shrivastava et al., 2017), where the SOA yield from VOCs is also uncertain.*

3-1: The use of the word "integrates" could be misconstrued as integration of the size distribution here. Consider a different word such as combines or includes.
**We have replaced with "assimilates".**
4-14: "as nearly as possible". Consider something like "as closely as possible".
4-19: The point of this sentence is valid, but these processes are not the only processes that control evolution of the particle size distribution.
**Replaced "process control the evolution" with "processes significantly affect the evolution".**

6-2: The terms here look somewhat like t_k(mu_k) as a function. Consider a change to something like "...depend on the scaling factors (mu_k), at time t_k ..." if appropriate.

7-1: Should mu_k_hat be referred to as "scaling factors" or similar here, as in 6-25 and 7-11, to prevent confusion?

**Yes, this has been changed.**

7-17: Even if restating from an earlier cited source, an additional reference for more information on the use of the condition number and relative sensitivity array in this methodology would be helpful for a reader wanting more information on this step.

**Citation to book by Highman, N. J., 2008 added.**

7-24: Consider "e.g." rather than "i.e." if appropriate.
8-25: Refer to inventory variables for each of these in this sentence, i.e. "...underpredicted aerosol mass (via dry aerosol volume, V_dry) as well as N_10..." or similar.

**These edits have been made.**

9-5: What was the justification for these ranges of rates spanning expected atmospherically relevant process rates?

**This is described in the sentence: "Since the objective is to design an inverse technique that is robust enough to apply in a global 3D CTM". We want to test the estimation technique on atmospherically relevant process rates to test the method's viability in a global CTM.**

Section 3.1: Was the uncertainty and estimated effect of instrument noise calculated using only the limited "meteorologically stable" time periods as opposed to the full year of data?

**No, as stated in the text the uncertainty is calculated only with the 23 well-conditioned synthetic measurements. We added clarification in the second sentence pointing to the measurements explored in the previous section (Section 3) at 10-24.**

10-16: Should be "damps" rather than "dampens".

**I think this is a personal preference as Merrian-Webster lists the verbs with the same definitions. Since "damps" can be considered a noun, adjective, or verb we will keep the less-ambiguous "dampens".**

13-15: Use of "integrate" again here. Consider "…way to combine information from …" or similar instead.
Fig 9: Needs a legend to show line color meaning in figure.

**These have been addressed in the revised manuscript.**

**RC2: 'Review of McGuffin et al. 2020'**

"In reality, emissions (and other processes) vary on a wide range of time scales, including seasonal, weekly, and diurnal cycles, and variability at shorter timescales. These variations are likely not fully captured by any CTM. Does this present a challenge for the choice of an appropriate value for the gain $K_c$? Can the authors comment on the robustness of their method to this issue?"

**The estimated scaling factor can be time-varying or constant depending on what the model needs to match the measurements, so the estimation method is already formulated to correct for timescales missing from the model. The most important decision to capture a missing timescale is defining an inventory variable that depicts that timescale in its measurements.**

"Could the authors discuss the sensitivity of the results to the sink rate chosen in this work? I understand that a detailed treatment of dilution, transport, and deposition is beyond the scope

of this study. However, given that the rates of these processes are both variable and uncertain, I think that a little further discussion is warranted. This discussion would also inform the potential of this method to be applied in a CTM, as there are uncertainties in more model processes than can be tested simultaneously using this method."

**In a CTM, the loss rate should be more representative than the consistent first-order rate used in this box model. However, if the modeled loss rate is slower than reality, i.e. not raining in the model, the estimation technique will not be able to match the model to the measurements. This is included in the main limitation of the estimation method – we assume all aerosol processes are known reasonably well except for nucleation, emissions, and growth.**

**We have added a paragraph discussing how this general issue of model uncertainties would affect the estimation performance at 7-29, which states:**

*Another drawback of this estimation method, which is shared with most inverse techniques, is the effect of uncertainty in model errors not corrected by the estimator ($f_k$ uncertainty). Model errors outside of particle nucleation, emissions, and growth can lead the estimator to over-compensate in the scaling factors if the inventory variables are sensitive to those model errors. For example, the estimation technique applied to a model that does not correctly simulate particle deposition will estimate particle emissions incorrectly while the estimated nucleation rate will not be affected. Identifying this scenario with outside model errors negatively impacting the estimated process rates is very difficult, but one sign is when the estimated process rates are not physical, e.g., a higher SOA production rate during nighttime. In future work, including various types of observations and more inventory variables can possibly inform the scaling factors and limit the impact of outside model errors on the estimated process rates.*

"I do not see the chosen values of the gain $K_c$ listed in the paper. It would be best to list them in section 2.2. It may be helpful to express them as convergence timescales."

**This is added at 7-1, which states:**

*For the rest of this paper, we use a matrix with diagonal elements of $[4, 4, 1]^T \ hr^{-1}$ for $K_c$, which corresponds to convergence timescales of 15 min and 1 hr for the number- and volume-based inventory variables, respectively. This array was found to have the best performance in reducing model-measurement mismatch without instabilities in the estimated scaling factors.*

"p4, line 20: Is it improved performance that the authors anticipate, or greater understanding? I would guess that the authors would find similar or better performance in using the mixing ratios directly as the control variables."

**We anticipate greater understanding; this has been revised.**

"p8, line 10: What is the timescale of the moving average? The authors later state that the synthetic noisy measurements are filtered with an 11-hour timescale. Was the same timescale applied to the observations?"

**The 11-hour window was used for noisy synthetic measurements (mentioned at 11-5), but without noise we used a window of 5 hours (added to text at 9-11). Then, for field measurements we used a window of 3 hours (added to text at 8-25) to smooth measurement noise without significantly dampening nucleation events.**

"p9, line 1, "will repeat": The authors should use the present tense here. Reserve the future tense for future work."

**This has been fixed in the text.**

"p10, line 31-33. I found this sentence confusing. If I am reading Fig. 7a correctly, the maximum in normalised mean bias increases from 0.06 to 0.09. Is the "maximum bias" the authors are referring to, then, the bias for a single time step of the box model (not shown in the figures)? If so, I would request that this be split into two sentences, as the second half does not refer to Fig. 7a."

**This sentence was confusing and not correct, so we have deleted it.**

"p5, line 1 states that the TOMAS model simulates particles as small as 0.5 nm in diameter. However, p12, line 31 seems to indicate that nucleated particles are generated with an initial diameter of 3 nm. Are the smallest size bins unused (i.e. always contain zero mass) in this study? This should be stated plainly in the methods section."

**This is correct, so we have added a statement on this at 5-1, which states:**

*Although TOMAS tracks the concentration of particles as small as 0.5 nm, the minimum predicted size is 3 nm here due to the implemented nucleation routine described below.*

"p13, lines 9-10: It may be clearer to say "this is after sunset on February 22 and before sunset on March 28th"."

**This has been edited.**

"Figure 7: Does the mean bias in the nucleation rate decrease when noise is added to the synthetic measurements? If so, this warrants a brief discussion in the text."

**No, there is not a significant change among the 23 scenarios, although the bias in a few cases may be lower with noise. We added the median, among the 23 scenarios, of the normalized mean bias to the text at 11-18:**

*In Figure 7a, we find that the normalized mean bias across the 23 scenarios does not significantly change with median values without and with noise, respectively, of 0.03 and 0.03, 0.005 and 0.007, and 0.004 and 0.006 for nucleation, emissions, and growth, respectively*

**RC3: 'Review of McGuffin et al.'**

"I believe the manuscript would benefit from further discussion on the limitations of applying this approach to ambient size distributions which may be influenced by many uncertain aerosol processes that are not being scaled in the inverse technique. As aerosol processes are often non-linear, how sensitive is this method to potential errors in the representation of other aerosol processes? The authors do introduce this issue in Section 4.2 (and I agree a full exploration of the problem is beyond the scope of this paper). What are the implications of the assumption that the other modeled aerosol processes are correct? If a given aerosol process is drastically misrepresented in the CTM, will this inverse approach overcompensate (attempting to get the correct answer for the wrong reason)?"

**You are correct that the estimation method will overcompensate for model errors outside of the scaled uncertain processes. We have added a paragraph discussing this issue at 7-29, which states:**

*Another drawback of this estimation method, which is shared with most inverse techniques, is the effect of uncertainty in model errors not corrected by the estimator ($f_k$ uncertainty). Model errors*

*outside of particle nucleation, emissions, and growth can lead the estimator to over-compensate in the scaling factors if the inventory variables are sensitive to those model errors. For example, the estimation technique applied to a model that does not correctly simulate particle deposition will estimate particle emissions incorrectly while the estimated nucleation rate will not be affected. Identifying this scenario with outside model errors negatively impacting the estimated process rates is very difficult, but one sign is when the estimated process rates are not physical, e.g., a higher SOA production rate during nighttime. In future work, including various types of observations and more inventory variables can possibly inform the scaling factors and limit the impact of outside model errors on the estimated process rates.*

"How generalizable is this approach in terms of choosing the scale factors and inventory variables? Would it be relatively straightforward for future studies to choose different aerosol processes to scale (for instance, if I wanted to assume nucleation rates are accurate but instead scale dry deposition rates)?"

**This approach is very generalizable as most of the difficulty is in choosing inventory variables that can capture the uncertain processes your model is missing. You can apply the same code we have implemented, but instead of calculating difference in your inventory variable before and after nucleation, you would do the difference before and after the aerosol process of interest, i.e., dry deposition.**

**In terms of choosing different scaling factors and inventories, you can choose any set you would like as long as you have at least as many inventories as scaling factors. Also, changes in the scaling factor must affect the predicted inventory variable to get good results. Ideally, you can choose an inventory variable in which the uncertain process is the dominating process.**

**We hope this is clearer in the text as we have slightly altered the text at 6-13 to read:**

*Because these rates are determined by finite difference, i.e., by saving model parameters before and after relevant subroutines are called, this approach is generalizable to various processes, modular, and robust with respect to changes in the aerosol microphysics. Internal details of subroutines can change so long as the estimator is able to compare the model state before and after the subroutine call. Additionally, this method can easily be adapted to other models with different uncertain processes or available measurements*

"How is the exponential error decay factor (Kc) tuned? Is it kept constant across the simulations using the synthetic and observed data or is it tuned in each simulation?"

**In the three sets of simulations performed, we used the same Kc. The values chosen were based on initial tests with the 27 scenarios of synthetic measurements so that the error converged to near-zero but also the estimated scaling factors did not become unstable. A short description of this is included in the text at 7-1, which states:**

*For the rest of this paper, we use a matrix with diagonal elements of $[4, 4, 1]^T \ hr^{-1}$ for $K_c$, which corresponds to convergence timescales of 15 min and 1 hr for the number- and volume-based inventory variables, respectively. This array was found to have the best performance in reducing model-measurement mismatch without instabilities in the estimated scaling factors.*

"Specific comments:

1. Line 2-13 could be rephrased as there are other processes that could contribute to aerosol growth not considered here."

**This has been changed to: *…particles grow partially due to…***

" 2. What is the normalized error for the aerosol properties simulated with the a priori TOMAS model in Figure 8? How does this compare when using the inverse method?"

**The *a priori* rates in TOMAS were chosen to be realistic, but we did not aim for this baseline model to match the measurements. Therefore, it is expected that the inverse method matches the inventory variables better than the baseline model with *a priori* rates. The normalized absolute error in the baseline simulation at the three measurement sites is tabulated here and added as a figure in the supplemental information (SI.1).**

| NAE: *A priori* simulation | N36 | N10 | Vdry |
|---|---|---|---|
| SPC | 0.98 | 0.48 | 0.38 |
| MPZ | 1.36 | 0.49 | 0.48 |
| HYY | 2.83 | 0.69 | 0.98 |

" 3. I think Figure 9 could benefit from a legend or additional annotation. I found it hard to remember each color representation."

**We added a legend to Figure 9.**